# Implications of Type 2 Diabetes Mellitus in Patients with Acute Cholangitis: A Systematic Review of Current Literature

**DOI:** 10.3390/healthcare10112196

**Published:** 2022-11-02

**Authors:** Matei-Alexandru Cozma, Elena-Codruta Dobrică, Purva Shah, Duha Shellah, Mihnea-Alexandru Găman, Camelia Cristina Diaconu

**Affiliations:** 1Faculty of Medicine, “Carol Davila” University of Medicine and Pharmacy, 050474 Bucharest, Romania; 2Department of Gastroenterology, Colentina Clinical Hospital, 020125 Bucharest, Romania; 3Faculty of Medicine, University of Medicine and Pharmacy of Craiova, 200349 Craiova, Romania; 4Department of Dermatology, “Elias” University Emergency Hospital, 011461 Bucharest, Romania; 5Research Associate, Department of Internal Medicine, Baroda Medical College, MS University, Vadodara 390001, India; 6Faculty of Medicine and Health Sciences, An-Najah National University, Nablus P.O. Box 7, Palestine; 7Department of Hematology, Center of Hematology and Bone Marrow Transplantation, Fundeni Clinical Institute, 022328 Bucharest, Romania; 8Department of Internal Medicine, Clinical Emergency Hospital of Bucharest, 14461 Bucharest, Romania

**Keywords:** acute cholangitis, diabetes mellitus, gallstone pancreatitis, ERCP

## Abstract

Introduction: Type 2 diabetes mellitus (T2DM) has been associated with higher rates and poorer prognosis of infections, mainly due to poor glycemic control, reduced response of T-cells and neutrophils, and impaired migration, phagocytosis, and chemotaxis of leukocytes. However, the impact of T2DM on acute cholangitis (AC) has not been assessed so far. Thus, we aimed to explore this association by means of a systematic review of the literature. Methods: This systematic review was carried out based on the recommendations stated in the Preferred Reporting Items for Systematic Reviews and Meta-Analyses (PRISMA) guidelines. We searched the PubMed/MEDLINE, Web of Science and SCOPUS databases to identify relevant publications depicting an association between T2DM and AC from the inception of these search services up to present. Results: We detected a total of 435 eligible records. After we applied the inclusion and exclusion criteria, a total of 14 articles were included in the present systematic review. Included manuscripts focused on the potential role of T2DM as a risk factor for the development of AC and on its contribution to a worse prognosis in AC, e.g., development of sepsis or other complications, the risk of AC recurrence and the impact on mortality. Conclusions: As compared to non-diabetic individuals, patients with T2DM have a higher risk of AC as a complication of choledocholithiasis or gallstone pancreatitis. Several oral hypoglycemic drugs used in the management of T2DM may also be involved in the onset of AC. Diabetic patients who suffer from AC have a higher likelihood of longer hospital stays and sepsis, as well as a higher risk of mortality and more severe forms of AC as compared to non-diabetic individuals.

## 1. Introduction

Biliary tract infections (BTI), which include acute cholecystitis and cholangitis (or angiocholitis), are relatively frequent and, in most cases, potentially life-threatening bacterial infections of the intra- and/or extrahepatic bile ducts [1,2,3,4]. They represent the second most common cause of community-acquired bacteremia and the third most common cause of hospital-acquired bacteremia, especially in elderly patients with pre-existing conditions, where rates can go as high as 70% [3,5,6,7]. Complications may develop in up to 20% of cases and remain a significant cause of mortality to this day, with rates ranging from 5 to 15%, despite advances in broad-spectrum antimicrobial therapy and improved access to emergency biliary tract decompression [3,6,7,8]. The pathophysiology of acute cholangitis (AC) is governed by two central phenomena, equally required for the development of cholangitis: bile flow restriction and bacterial colonization and proliferation in the biliary tract [9,10,11,12]. Common causes of biliary obstructions are gallstones or biliary duct stenoses, usually caused by chronic pancreatitis, malignant or benign proliferations, or sclerosing cholangitis [4,10,13]. Normally, the bile and the biliary tract are sterile. Yet bacterial colonization of the biliary system has no clinical importance under conditions of normal bile flow. However, when bile duct obstruction occurs, bacteria proliferate within the stagnant bile while biliary pressures increase. Eventually, bacteria presumably translocate into the circulation causing a systemic infection, which can lead to sepsis and multiple systems organ failure [12,14]. Historically, the diagnosis was clinical, using Charcot’s triad (right upper quadrant pain, fever/chills, and jaundice). While it has a very high specificity (96%) and lack of sensitivity (26%), because only a third of patients present with Charcot’s triad, diagnosis is now made based on Tokyo Guidelines (TG) proposed by the Tokyo consensus for the first time in 2007 and the revisions in 2013 and 2018 [10,15]. These new criteria include clinical or laboratory signs of systemic inflammation, cholestasis, and imaging evidence of biliary obstruction and have an overall sensitivity and specificity scores of 92% and 78%, respectively [10,15,16].

Early biliary decompression and rapid initiation of empiric broad-spectrum antibiotics represent the mainstays of treatment for AC. The restoration of the bile flow can be achieved surgically, percutaneously, or by endoscopy, but the 2018 TG for the management of AC recommend early endoscopic retrograde cholangiopancreatography (ERCP) (due to the minimal invasiveness of this procedure) to be performed within 48 or 72 h [3,6,17]. Because microbiological identification of pathogens requires time, the same consensus recommends an initial empiric course of antibiotics, typically with third-generation cephalosporins or a penicillin/beta-lactamase inhibitor-based agent to be initiated in the first hour of diagnosis [4,6,18]. T2DM is considered one of the largest emerging threats to health in the 21st century, being the seventh leading cause of death across the globe [19,20,21]. The number of people suffering from T2DM is projected to rise from 415 million, as estimated in 2015, to over 640 million in 2040, affecting approximately 1 in 3 Americans [19,20,22]. Besides the well-known vascular and neurological complications, it is sure now that T2DM also increases the risk of infections, causing almost 6% of the patients to present to the hospital [19,20,23]. The mechanism behind this particularly high susceptibility is not very well-understood, but it is thought to be secondary to poor glycemic control, diabetic neuropathy, reduced response of T-cells and neutrophils, and impaired migration, phagocytosis, intracellular killing, and chemotaxis of leukocytes [20,21,23].

Although diabetic patients can also develop AC, the impact of T2DM on AC development and outcome has not been assessed so far. Thus, the aim of this systematic review was to provide an overview of the implications of T2DM in patients with AC, namely the risk of readmission, the severity of AC, or the risk of AC complications. Although it is well known that patients with T2DM are prone to acquire infections of any kind, which complicates the management of this cardiometabolic disorder, no systematic review or meta-analysis have been conducted so far to identify and analyze the impact of T2DM on AC. 

## 2. Materials and Methods

### Data Sources and Searches

This manuscript was prepared following the recommendations mentioned in the PRISMA guidelines [24]. Three independent investigators (M.-A.C., E.-C.D., M.-A.G.) performed an advanced search in 3 databases (PubMed/Medline, Web of Science, and SCOPUS) to retrieve relevant studies published from the inception of these databases and up to 1 June 2022. The search was performed using specific keyword and/or word combinations: (cholangitis[MeSH] OR “acute cholangitis” OR “bile ducts inflammation” OR “angiocholitis” OR “choledochitis”) AND (“diabetes*” OR diabetes[MeSH]). All potentially valid studies resulting from the first scan were considered for this systematic review. 

The following inclusion criteria were taken into consideration for the final selection of the articles: 1. Original studies which evaluated the prognosis, risk factors, outcomes, or management strategies of cholangitis and its complications OR original studies which evaluated the relationship between cholangitis and diabetes. 2. Studies conducted on the adult population (subjects with an age equal to or higher than 18 years old). 3. The included articles were written in English, French, Italian, or Romanian (the languages spoken by the main investigators of the review). 4. The publication year of the included articles was after 1 January 2000. Exclusion criteria were as follows: 1. Studies performed on a population with subjects aged less than 18 years old, on cell cultures, or on animals. 2. Manuscripts written in a language that was not spoken by the authors. 3. Reviews, letters to editor, case reports, or abstracts presented at scientific events. 4. Studies published before 1 January 2000. 5. Studies that did not show any relationship between diabetes and cholangitis or any impact of this comorbidity on the prognosis, management, or treatment.

All the articles included in the study met the inclusion and exclusion criteria as agreed unanimously by all the investigators. Any disagreement between the investigators was resolved by consultation with the senior author (C.C.D.) to allow for the final selection of the papers to be included in this systematic review. We evaluated the methodological quality and the risk of bias of the analyzed studies using the Methodological Index for Non-Randomized Observational Studies (MINORS) and the Mixed Methods Appraisal Tool (MMAT), respectively [25,26]. This protocol was registered in PROSPERO (ID 338250). No ethical approval or letter of informed consent were required to carry out this research, and all the files from the articles taken into account are retrievable from the searched databases.

## 3. Results

The flow-chart diagram of the literature search process is reported in Figure 1. We detected a total of 435 eligible records. Of these, after we removed the duplicates and the papers excluded after the screening of titles and abstracts step (*n* = 231), 204 original articles were selected for full-text review. After we applied the inclusion and exclusion criteria, a total of 14 articles were included in the present systematic review. 

The 14 selected manuscripts were published between 2011 and 2022 [27,28,29,30,31,32,33,34,35,36,37,38,39,40]. Most of the analyzed data were derived from retrospective cohort studies [27,28,29,30,31,32,33,34,35,37,38,39,40] with the exception of two investigations, namely one prospective study [36] and one case series [34]. The number of subjects who partook in these assessments ranged from 12 to 123,990 [27,28,29,30,31,32,33,34,35,36,37,38,39,40]. Most of the research was carried out in Asia: two studies were conducted in Singapore [28,36], two in China [31,32], one in India [37], one in South Korea [35], and one in Israel [33]. The remaining studies were conducted in the United States of America, i.e., two studies [27,38], and in Europe, i.e., Denmark [40], France [34], or Spain [30], or were the fruit of international collaboration [39]. Most manuscripts focused on the potential role of DM as a risk factor for the development of AC or on its contribution to a worse prognosis (e.g., sepsis, recurrence) or to an elevated mortality in AC [27,28,29,30,31,32,33,35,36,38,40]. Two studies focused on the etiology of AC and sought to depict an association between the micro-organisms involved and the presence of DM [34,37]. One assessment pointed out an association between antidiabetic drugs and the risk of AC [39]. The main particularities identified in this systematic review are summarized in Table 1.

Overall, T2DM increased the risk of AC onset in patients with choledocholithiasis, gallstone pancreatitis, patients who underwent ERCP, and subjects with T2DM who were treated with oral hypoglycemic agents. T2DM was associated with longer hospital stay, sepsis, higher risk of mortality, and more severe forms of AC as compared to non-diabetic individuals [27,28,29,30,31,32,33,34,35,36,37,38,39,40].

## 4. Discussion

This systematic review focused on the impact of T2DM in patients with AC and discovered that T2DM is associated with worse outcomes in AC patients. T2DM also increased the risk of AC onset in patients with choledocholithiasis, gallstone pancreatitis, patients who underwent ERCP, and patients on oral hypoglycemic treatment for T2DM. T2DM was associated with a longer hospital stay, sepsis, higher risk of mortality, and more severe AC as compared to non-diabetic patients. Figure 2 depicts the implications of the T2DM and AC association [27,28,29,30,31,32,33,34,35,36,37,38,39,40].

### 4.1. Research on T2DM and AC: Why Is There Still a Need for This Review?

As per our search in the explored databases, we were not able to identify any systematic reviews or meta-analyses on the association of T2DM and the outcome of AC. We believe this systematic review provides valuable information about how the changes in severity, mortality risk, length of hospital stay, and risk of bacteremia in AC are associated with T2DM. In our systematic review, 14 studies were included. Eight studies were focused on predicting bile duct complications in individuals with AC [27,28,32,33,34,35,39,40]. Six studies concentrated on predicting complications in the treatment of AC [29,30,31,36,37,38].

### 4.2. Mechanism of Acute Cholangitis

Common bile duct (CBD) blockage due to choledocholithiasis, sclerosing cholangitis, malignancy, and bile duct instrumentation superimposed by intraductal bacterial overgrowth causes AC. Increased intraductal pressure secondary to CBD blockage is transmitted to biliary ductules, which increases their permeability to bacteria and bacterial toxins. Entry of pathogens and their products into the blood stream or surrounding tissues leads to complications, e.g., sepsis and hepatic abscess [41,42,43,44]. Hence, early diagnosis and treatment of AC is important to avoid serious complications and death.

#### 4.2.1. Risk Factors for AC

Based on the analyzed data, we identified the following risk factors for suppurative AC: age above 70 years old, being a smoker, impacted duct stones, and the presence of gallstones [41]. Renal dysfunction, choledocholithiasis, and the identification of extended spectrum beta-lactamase-producing micro-organisms as the source of infection emerged as risk factors for organ failure in patients with AC [41]. Moreover, we discovered that no percutaneous cholecystostomy, insufficient drainage, mental confusion, hypotension-requiring catecholamines, organ failure, leukocytosis, hyperbilirubinemia, Quick’s value, high serum creatinine, bacteremia, and thrombocytopenia were risk factors for mortality in AC (mortality risk > 0.7%).

#### 4.2.2. Association of AC and T2DM

T2DM is known to cause dyslipidemia, as evident by the higher levels of triglycerides (TG), intermediate-density lipoproteins (IDL), very low-density lipoproteins (VLDL), and the lower concentrations of high-density lipoproteins (HDL) in individuals with T2DM as compared to subjects without this cardiometabolic disorder. While low-density lipoprotein (LDL) levels remain the same in T2DM, the levels of small dense LDL particles increase [45]. This leads to the production of lithogenic bile, which is highly saturated with cholesterol and consequently increases the risk of gallstones in T2DM [46].

Several investigations have highlighted that T2DM promotes bacterial proliferation in the bile ducts and also increases the risk of infected gallstones. As a result of diabetic neuropathy and angiopathy, blood flow to the bile ducts and gallbladder reduces, along with lower gallbladder contractility. This leads to bile stasis and increases the risk of biliary infections [47]. Gallstones in patients with T2DM show bacterial exotoxins, DNA, and proteins in a higher proportion as compared to non-diabetic subjects [48].

Autonomic neuropathy in T2DM also causes the sphincter of Oddi dysfunction resulting in a high resting tone in fasting states, promoting cholestasis. Gastrointestinal dyskinesia in T2DM elevates gut anaerobic bacteria and gut pH, which hastens the production of deoxycholic acid. Higher levels of hydrophobic bile acids through the enterohepatic circulation lead to crystallization of cholesterol, hence leading to gallstones [47].

### 4.3. Predicting Bile Duct Complications

Choledocholithiasis complicated by AC was highlighted in two studies [27,33]. A retrospective study by Kummerow et al. evaluated, in a group of 123,990 patients discharged with the diagnosis of choledocholithiasis, the risk factors for evolution into a complicated form (with acute pancreatitis or associated cholangitis), as well as predictive factors associated with higher mortality. Thus, it was found that 17.2% of patients with acute pancreatitis and 21.8% of patients with cholangitis have diabetes compared to only 15% of those with uncomplicated choledocholithiasis (OR = 1.1, CI = 1–1.2). Moreover, it was found that the presence of diabetes is the third predictor of mortality (OR = 1.14), after the presence of complications and alcoholism [27].

One of the first studies which demonstrated the relationship between diabetes and AC in patients with common bile duct stones (CBDS) was the one developed by Khoury and Sbeit, which compared 101 patients with AC and CBDS (mean age 77.7 ± 13.6 years, 47.5% females) and 586 patients without AC and CBDS (mean age 62.5 ± 20.5 years, 58.4% females). They found that the presence of T2DM was more frequently associated with AC development, the association being statistically significant (OR = 1.92, *p* = 0.002) [33].

### 4.4. Assessing the Association of T2DM with Gallstone Pancreatitis and Bile Duct Complications

In a study conducted by Kim et al. on 290 patients with acute biliary pancreatitis (mean age 66.8 ± 16.0 years, 53.1% female), a higher number of bile duct-related complications (acute pancreatitis, cholecystitis, or cholangitis) occurred in patients with diabetes compared to those without this comorbidity, but the difference was not statistically significant (*p* = 0.33) [35]. Another study conducted by Charlier et al. evaluated the characteristics of patients with *Listeria monocytogenes* infection of the biliary tract, which is an infection that develops mostly in patients with an immunocompromised status. The study evaluated 20 patients (50% women, mean age 69 years) of whom only 4 associated diabetes as a comorbidity [34].

### 4.5. Post-ERCP Bile Duct Complications

The risk of cholangitis after ERCP is well-known, and multiple studies have delineated different risk factors for post-ERCP complications. In a retrospective study that included 110 patients (50% females, 74 aged >60 years) who were treated by ERCP and developed cholangitis and 2174 patients who were subjected to ERCP but did not experience any complication, T2DM emerged as a risk factor for post-ERCP complications (*p* < 0.05). Other risk factors for complications mentioned in the study which were statistically significant were age, hypertension, previous ERCP, pancreatography, interventional placed biliary stent, balloon dilatation techniques, obstruction on different sites of ducts, and calculus extraction by endoscopic techniques [32]. Chen et al. performed a retrospective single-center cohort study consisting of 4234 patients who had undergone ERCP and found that, among other factors such as hypertension, previous history of ERCP, pancreatography, sphincterotomy, and balloon dilatation, T2DM was associated with a higher risk of developing post-ERCP complications (OR = 0.527, 95% CI = 0.274–1.014) [32].

### 4.6. T2DM Management Complicated by AC

Due to stimulation of cholangiocytes′ proliferation by GLP-1, GLP-1 analogues and DPP-4 inhibitors used in the management of T2DM increase the risk of gallbladder and bile duct diseases such as cholangitis, cholecystitis, choledocholithiasis, and gallbladder cancer [39]. Faillie et al.’s (2016) comparative study on 71,369 T2DM patients, of whom 853 were hospitalized for bile duct and gallbladder disease (incidence rate per 1000 person-years, 3.7; 95% CI, 3.5–4.0), assessed the association of the use of GLP-1 analogues and DPP-4 inhibitors and the increased risk of bile duct and gallbladder diseases. Their results show that GLP-1 use was associated with a 79% elevated risk of bile duct and gallbladder diseases compared with current use of at least two oral antidiabetic drugs (6.1 vs. 3.3 per 1000 person-years; adjusted HR, 1.79; 95% CI, 1.21–2.67). However, there was no association between using DPP-4 inhibitors and bile duct and gallbladder diseases compared with current use of at least two oral antidiabetic drugs (3.6 vs. 3.3 per 1000 person-years; adjusted HR, 0.99; 95% CI, 0.75–1.32). In a secondary analysis, GLP-1 analogues were also associated with an increased risk of cholecystectomy (adjusted HR, 2.08; 95% CI) [39].

### 4.7. Predicting the Severity of AC and the Risk of Mortality

Mohan et al., in a retrospective study of 388 patients diagnosed with AC between January 2009 and December 2016, looked at establishing predictive factors for in-hospital mortality. Thus, T2DM was diagnosed in 46.8% of those with a definite diagnosis, compared to 39.6% of those with an uncertain diagnosis and 38% of those with an excluded diagnosis, without having a statistically significant value. In addition, diabetes was present in a higher proportion in severe forms (48%), compared to moderate (37.8%) and mild (40.3%) ones (*p* = 0.19) [28]. In Tan et al.’s (2019) epidemiological study in Denmark on 755 patients with cholangitis, 176 were diabetic and 29 of them experienced complications. Among the 755 cases, 42% (*n*  =  326) were of malignant etiology, with an increasing incidence over time (regression coefficient [95% CI]: 0.03 [0.01–0.04] per year; *p* = 0.01). The average value of the Charlson Comorbidity Index was 1.4, with an increase over time (regression coefficient [95% CI]: 0.04 [0.03–0.05] per year; *p* <  0.01). Malignant obstruction etiology was associated with 30-day mortality (OR [95% CI]: 1.11 [1.04–1.18]; *p* < 0.01). Overall, 30-day mortality was 12% (*n*  =  91). After adjustment for confounding factors, no significant changes in 30-day mortality were observed over time (OR [95% CI]: 1 [1–1.00]; *p* = 0.91 per year) [40].

### 4.8. Predicting Complications in the Treatment of AC

#### 4.8.1. Non-Surgical Approach for AC in T2DM

Another retrospective study by Garcia-Alonso et al. (491 patients, mean age = 78.8 years (71.9–84.7 years), 51.7% women, 117 patients with diabetes), which evaluated the risk factors for cholecystectomy after an acute event (acute pancreatitis, cholangitis, cholecystitis), underlined that diabetes was associated more frequently with a non-surgical approach (*p* = 0.09) [30].

#### 4.8.2. Predicting Complications during AC Management in T2DM

Diabetes seems to also be an important predictive factor for sepsis development after AC. Liu et al. created a predictive model for the risk of sepsis in which diabetes (OR = 10.74, 95% CI = 2.80–70.57, *p* < 0.01) was an important variable together with age, ventilator-support time, systolic blood pressure, and coagulopathy [31]. High prevalence of cardiovascular events and renal failure in T2DM increase postoperative mortality after emergent cholecystectomy in diabetic AC patients as compared to non-diabetics [49].

#### 4.8.3. Predicting Increased Length of Hospital Stay

A prospective study conducted by Mak et al. in Singapore on 124 patients (median age 64.5 years, 39.5% female) with hepatobiliary infections evaluated the presence of factors associated with admission in high-dependency units (HDU) or prolonged hospital stay. They underlined that diabetes was an independent predictive factor on multivariable analysis (*p* = 0.003) for severe cases with admission in HDU. In addition, diabetes was the second most encountered comorbidity in those patients (34.7%) after hypertension (55.6%) [36].

#### 4.8.4. Predicting Antibiotic Resistance

T2DM patients have a higher risk of contracting resistant infections because of an impaired immune system and frequent hospital visits. Obesity, which is closely associated to T2DM, leads to high proinflammatory cytokines (IL1, IL6, IL8, TNF-alpha) and adipokines which further hinder the subjects’ immunity. Due to a high BMI and fat distribution in obese patients with T2DM, blood concentrations of antibiotics remain suboptimal, elevating the risk of antibiotic resistance in this patient population [50]. Addressing the need for a change in empirical antibiotic policy for AC patients due to a shift in the antibiotic susceptibility of causative organisms, Sahu et al. found that out of 185 patients (median age 51.3 years, 55.1% males), 25 (13.5%) patients had pre-existing T2DM. Choledocholithiasis and malignancy were the most common instigating factors of gallbladder inflammation. The previously used ampicillin–gentamicin combination seemed to be less useful due to rising resistance against ampicillin and higher nephrotoxic effects of gentamicin [37].

#### 4.8.5. Aiming to Predict the Risk of Recurrence and Readmission after Treatment of AC

A retrospective cohort study conducted on 182 patients (median age 64 years; 49% females) by Jirapinyo et al. found that stent exchange at the time of AC was more effective in preventing recurrence than stent sweeping or the stent-in-stent approach. Thirty percent of the study participants had a history of T2DM, and absence of T2DM did not decrease the risk of disease recurrence. T2DM is a risk factor for recurrent bouts of AC; hence, caution must be practiced while using stents that are not designed for removal—for example, uncovered metal stents [38]. Regarding the risk of readmission 30 days after, Parikh et al. underlined that the risk is significantly greater in those with diabetes (*p* = 0.003) than in the patients without this comorbidity. Thus, T2DM is a risk factor for readmission of patients with AC, together with age, morbid obesity, and other choledocian complications/interventions [29].

### 4.9. Recommendations for Future Studies

The authors of this review recommend that future clinical trials on this topic be conducted with larger sample sizes and comparison or control arms. We would also recommend the use of TG for risk and severity stratification criteria for AC [28]. Quantitative analysis may provide stronger scientific value to papers as compared to qualitative analysis conducted in systematic reviews.

## 5. Conclusions

As compared to non-diabetic individuals, patients with T2DM have a higher risk of AC as a complication of choledocholithiasis, gallstone pancreatitis, or ERCP. Some oral hypoglycemic drugs used in the management of T2DM may also cause AC. Diabetic patients suffering from AC have a higher likelihood of a longer hospital stay, sepsis, elevated risk of mortality, and more severe forms of AC as compared to non-diabetic individuals. Hence, this systematic review found an association between T2DM and worse outcomes in patients suffering from AC. The statistical significance of these findings may be found by performing quantitative analysis in future studies. Given the small number and the limitations of the studies included in this systematic review, the authors recommend that further large prospective studies are needed to investigate how not only T2DM as a whole, but, more importantly, its duration, glycemic control, and other components of the metabolic syndrome, influence the prognosis of patients with AC. A comparison or control arm and the use of the 2018 TG for risk and severity stratification criteria for AC should also be used to further increase the quality of the results. Last but not least, it is equally important to identify which are the best management options for these particular patients in order to achieve lower mortality and incidence of complications. Quantitative analysis may provide a stronger scientific value to papers as compared to qualitative analysis conducted in systematic reviews.

## Figures and Tables

**Figure 1 healthcare-10-02196-f001:**
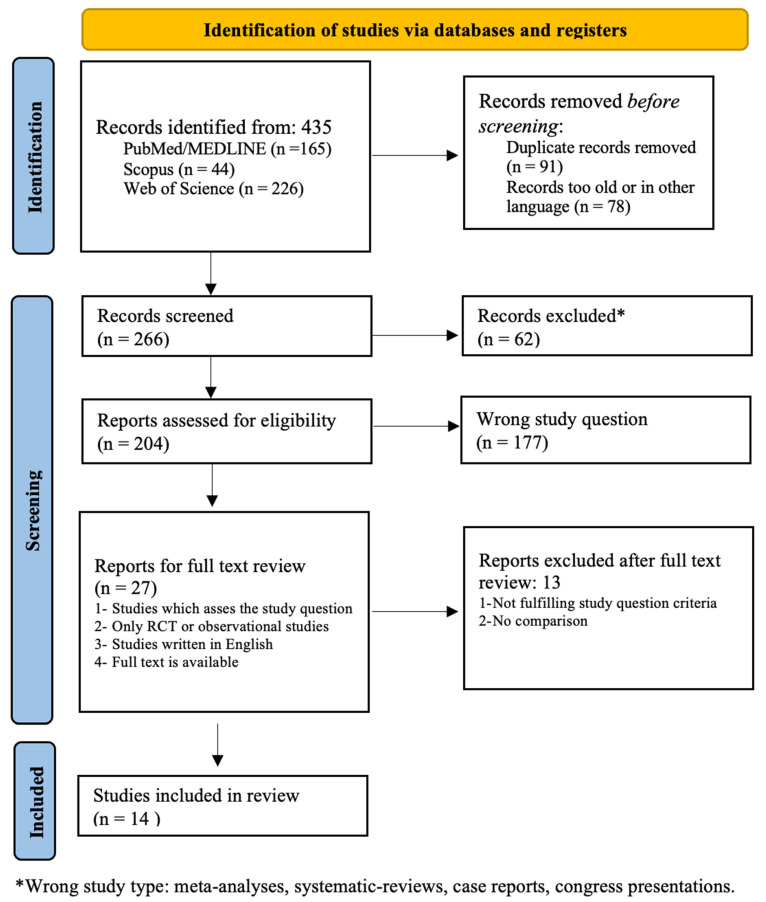
PRISMA 2020 flow diagram. From: Page MJ, McKenzie JE, Bossuyt PM, Boutron I, Hoffmann TC, Mulrow CD, et al. The PRISMA 2020 statement: an updated guideline for reporting systematic reviews. BMJ 2021;372:n71. doi: 10.1136/bmj.n71. For more information, visit: http://www.prisma-statement.org/ (accessed on 15 June 2022) [24].

**Figure 2 healthcare-10-02196-f002:**
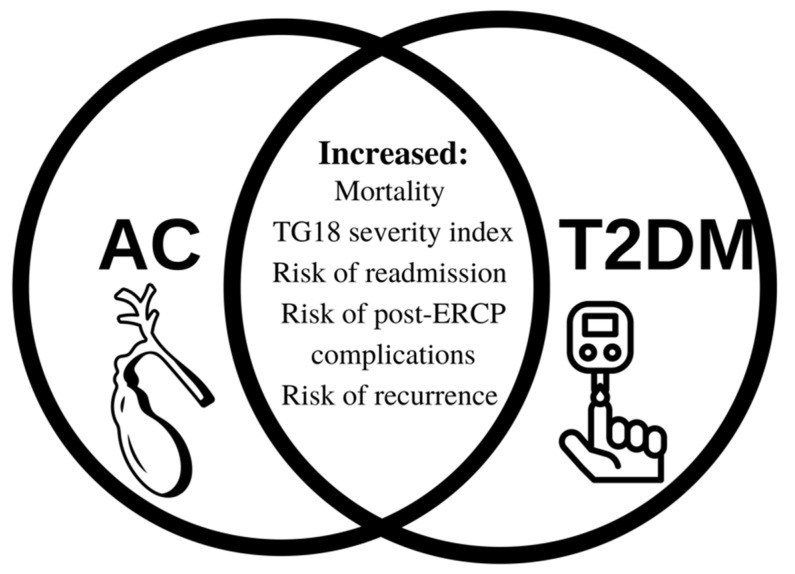
The main implications of type 2 diabetes mellitus in patients with acute cholangitis. Legend: AC, acute cholangitis. T2DM, type 2 diabetes mellitus. TG18, Tokyo Guidelines 2018. ERCP, endoscopic retrograde cholangiopancreatography.

**Table 1 healthcare-10-02196-t001:** Main particularities identified in diabetic patients suffering from acute cholangitis. AC—acute cholangitis, T2DM—type 2 diabetes mellitus, AMI—acute myocardial infarction, CVA—cerebral vascular accident, CHF—congestive heart failure, PUD—peptic ulcer disease, PVD—peripheral vascular disease, HT—hypertension, AP—acute pancreatitis, AoV—ampulla of vater, CBD—common bile duct, pst-patients, CL—cholecystitis.

First Author	Year of Publication	Study Design	Country	No. of Participants	Mean Age	Gender	No. of Participants with T2DM	No. of Participants without T2DM	Outcome of AC in pts with DM	Outcome of AC in pts without T2DM	Other Comorbidities
Kummerow et al. [27]	2012	Retrospective study	USA	123,990	Uncomplicated CL 56.5 ± 0.36 year AP-60.3 0.40 y	Male/Female	21.8% of patients with AC had T2DM—20,956	20,956	Third predictor of mortality (OR = 1.14), after complications and alcoholism.		CHF, chronic lung disease, T2DM, HT, obesity, renal failure
Mohan et al. [28]	2021	Retrospective study	Singapore	388	75.9 years	Male/Female	Diagnosed in 46.8% of those with a definite diagnosis—100	162	Higher proportion in severe forms (48%), compared to moderate (37.8%) and mild (40.3%) ones (*p* = 0.197).		
Parikh et al. [29]	2021	Retrospective cohort study	USA			Male/Female			Risk of readmission at 30 days is significantly greater in those with T2DM (*p* = 0.003).		
Garcia-Alonso et al. [30]	2015	Retrospective study		491	78.8 (71.9–84.7)	51.7% women	117	374	After an acute event (AP, AC, cholecystitis) underlined that diabetes was associated more frequent with a non-surgical approach (*p* = 0.09).		
Liu et al. [31]	2020	Retrospective review	USA	662	70.7 +/− 14.7	Male/Female	23	639	Diabetes (OR: 10.74, 95% CI: 2.80–70.57) was associated with the risk of sepsis in AC.		
Chen M et al. [32]	2018	Retrospective study	China	110 + 2174	74 over 60 years old	50% females	256	3978	Risk factor for post-ERCP complications such as AC (*p* < 0.05).		
Khoury and Sbeit [33]	2022	Retrospective study	Israel	101 + 586	77.7 ± 13.6	47.5% females	53 (52.5) in group A (CL with AC) and 213 (36.3) in group B (CL without AC) (*p* = 0.001)	48 in group A and 373 in group B	Associated with AC development (OR 1.92, *p* = 0.002).		
Charlier et al. [34]	2014	Retrospective study	France	20	69	50% female	3	17	Recurrence in 1 patient needing 6 weeks amoxicillin and 2 drainages f/b cure in 2 yrs. Cure in other 2 pts.		
Kim et al. [35]	2017	Retrospective study	Republic of Korea	290	66.8 ± 16.0	53.1% female	52	238	CBD-related complications (AP, CL or AC) (*p* = 0.334).		
Mak et al. [36]	2019	Prospective study	Singapore	124	64.5	39.5% female	34 (7%)	90	Severe cases with admission in high-dependency units (HDU) (*p* = 0.003).		
Sahu MK et al. [37]	2011	Retrospective and prospective study	India	185	51.3 ± 13.4	55.1% male	25 (13.5%)	160	Not mentioned as paper is about need for a change in empirical antibiotic policy for AC.	Not mentioned.	
Jirapinyo et al. [38]	2019	Retrospective cohort	USA	182	64	49% female	55 (30%)	127	Not mentioned as paper is about stent exchange at the time of AC, which was more effective in preventing recurrence.	Absence of T2DM did not decrease the risk of disease recurrence.	HT
Faillie J et al. [39]	2016	Comparative study	Canada	71369	GLP-1 analogous 57.1 years DPP-4 inhibitors 65.1 years Others 62.3 years	Male/Female	71369	None	This study assessed AC and other gallbladder diseases as a complication from noninsulin antidiabetic drugs, mainly: GLP-1 analogues and DPP-4 inhibitors not as a complication from DM itself. Total of 853 were hospitalized for gallbladder disease.	Null	Null
Tan M et al. [40]	2019	Epidemiological study	Denmark	755	72 ± 9	Male/Female	176 (29 had complications)	579	All of them having AC.	All of them having AC.	AMI, cancer, CVA, CHF, connective tissue disorder, dementia, HIV 1, liver disease, metastatic cancer, paraplegia, PUD, PVD, pulmonary disease, renal disease, severe liver disease.

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
