# Peer review of "Implications of Type 2 Diabetes Mellitus in Patients with Acute Cholangitis: A Systematic Review of Current Literature"

_healthcare, 2022, doi:10.3390/healthcare10112196_

Round 1

Reviewer 1 Report

1. Your major objective and approaches to discuss this systematic review should be elaborated in the last paragraph of the introduction section.  2. Discussion section needs to be elaborated with recent references.  3. Conclusion should be more about the future prospective of this systematic review along with areas left for further study.  4. A signalling figure as per the title and conclusion of your manuscript is needed to improve the quality of the manuscript.  5. Complete editorial checking will be needed to correct the grammatical and punctuation mistakes. 

Author Response

Dear Academic Editor,

Dear Peer-Reviewers,

We are very thankful to you and to the peer-reviewers for the pertinent notes; we have carefully read the comments and have revised/completed the manuscript accordingly. Our responses are given in a point-by-point manner below. All the changes to the manuscript are highlighted in yellow.

We hope that, in this new form, the manuscript will be suitable for publication in Healthcare.

Reviewer 1

Response: We would like to thank you for your valuable comments which helped us improve the manuscript. All suggestions were taken into consideration and appropriate information, as well as required corrections, were provided. New/corrected parts are highlighted in yellow to facilitate the assessment of changes. We did our best to fulfil the expectations and we hope that you will be satisfied with our corrections.

All in all, we thank you for your positive comments and appreciation regarding our manuscript.

  1. Your major objective and approaches to discuss this systematic review should be elaborated in the last paragraph of the introduction section.

Response: Thank you for your valuable suggestion. We have amended the last paragraph of the introduction accordingly: Although diabetic patients can also develop AC, the impact of T2DM on AC de-velopment and outcome has not been assessed so far. Thus, the aim of this systematic review was to provide an overview of the implications of T2DM in patients with AC, namely the risk of readmission, the severity of AC or the risk of AC complications. Although it is well known that patients with T2DM are prone to acquire infections of any kind which complicates the management of this cardiometabolic disorder, no sys-tematic review or meta-analysis have been conducted so far to identify and analyze the impact of T2DM on AC.

  1. Discussion section needs to be elaborated with recent references.

Response: Thank you for your valuable suggestions. We have included a couple of new references to the Discussions. However, we must point out that not a lot of articles have focused on the association between diabetes and acute cholangitis. This explains why our review aimed to explore this topic.

  1. Conclusion should be more about the future prospective of this systematic review along with areas left for further study

Response: Thank you for your valuable suggestion. We have amended the conclusions in agreement with your comments.

As compared to non-diabetic individuals, patients with T2DM have a higher risk of AC as a complication of choledocholithiasis, gallstone pancreatitis, or ERCP. Some oral hypoglycemic drugs used in the management of T2DM may also cause AC. Dia-betic patients suffering from AC have a higher likelihood of a longer hospital stay, sep-sis, elevated risk of mortality, and more severe forms of AC as compared to non-diabetic individuals. Hence, this systematic review found an association between T2DM and worse outcomes in patients suffering from AC. Statistical significance of these find-ings may be found by performing quantitative analysis in future studies. Given the small number and the limitations of the studies included in this systematic review, the authors recommend further large prospective studies are needed to investigate how, not only T2DM as a whole but, more importantly, its duration, glycemic control and other components of the metabolic syndrome influence the prognosis of patients with AC. A comparison or control arm and the use of the 2018 TG for risk and severity strat-ification criteria for AC should also be used to further increase the quality of the results. Last but not least, it is equally important to identify which are the best management options for these particular patients in order to achieve lower mortality and incidence of complications. Quantitative analysis may provide a stronger scientific value to papers as compared to qualitative analysis conducted in systematic reviews. 4. A signalling figure as per the title and conclusion of your manuscript is needed to improve the quality of the manuscript.

Response: Thank you for your valuable suggestion. We have designed an original figure to capture the main aspects of our paper.

  1. Complete editorial checking will be needed to correct the grammatical and punctuation mistakes.

Response: Thank you for your valuable suggestion. The scientific writing and English language of the manuscript was reviewed by a native speaker of English certified by the University of Cambridge, UK. In addition, we have point out that, if accepted, any minor grammatical and punctuation errors will be corrected during the proofreading stage.

Thank you for your valuable suggestions. We do hope you will find the revised version of the paper significantly improved and worthy of publication in Healthcare.

Yours sincerely,

The authors.

Reviewer 2 Report

Dr Matei-Alexandru Cozma and co-authors present a review on the association between T2DM and acute cholangitis. The idea is intriguing, however, in my opinion, the manuscript lacks a solid methodology and analysis supporting the authors’ conclusions. Additional comments are provided below:

 Abstract:

The abstract should be divided in the following four sections Introduction - Methods - Results – Conclusions. The section results should report objective and precise data on the analysis performed. Conclusions must be added to the abstract.

 Methods:

-          “Full-text access was possible for the included studies” should not be an inclusion/exclusion criterion. Please specify how many full-text you excluded as these cannot be retrieved and the specific reasons why those full-text could not be retrieved.

Results:

-          The results section results are poor. No analysis has been carried out. Data has not been summarised in the results sections. It should be definitively edited. Some findings are described in the discussion sections “T2DM was associated with a longer hospital stay, sepsis, higher risk of mortality, and more severe AC as compared to non-diabetic patients”. However, these findings are not supported by any analytical effort provided by the authors.

 Conclusions:

-          Conclusions are not supported by data provided by the present paper.

 Minor:

There are some typos throughout the text (Web of Science in fig. 1)

Author Response

Dear Academic Editor,

Dear Peer-Reviewers,

We are very thankful to you and to the peer-reviewers for the pertinent notes; we have carefully read the comments and have revised/completed the manuscript accordingly. Our responses are given in a point-by-point manner below. All the changes to the manuscript are highlighted in yellow.

We hope that, in this new form, the manuscript will be suitable for publication in Healthcare.

Reviewer 2

Dr Matei-Alexandru Cozma and co-authors present a review on the association between T2DM and acute cholangitis. The idea is intriguing, however, in my opinion, the manuscript lacks a solid methodology and analysis supporting the authors’ conclusions. Additional comments are provided below:

Response: We would like to thank you for your valuable comments which helped us improve the manuscript. All suggestions were taken into consideration and appropriate information, as well as required corrections, were provided. New/corrected parts are highlighted in yellow to facilitate the assessment of changes. We did our best to fulfil the expectations and we hope that you will be satisfied with our corrections.

All in all, we thank you for your positive comments and appreciation regarding our manuscript.

 Abstract: The abstract should be divided in the following four sections Introduction - Methods - Results – Conclusions. The section results should report objective and precise data on the analysis performed. Conclusions must be added to the abstract.

Response: Thank you for your valuable suggestion. We have revised the abstract in accordance with your suggestion.

Abstract: Introduction: Type 2 diabetes mellitus (T2DM) has been associated with higher rates and poorer prognosis of infections, mainly due to poor glycemic control, reduced response of T-cells and neutrophils, and impaired migration, phagocytosis and chemotaxis of leukocytes. How-ever, the impact of T2DM on acute cholangitis (AC) has not been assessed so far. Thus, we aimed to explore this association by means of a systematic review of the literature. Methods: This sys-tematic review was carried out based on the recommendations stated in the Preferred Reporting Items for Systematic Reviews and Meta-Analyses (PRISMA) guidelines. We searched the Pub-Med/MEDLINE, Web of Science and SCOPUS databases to identify relevant publications depict-ing an association between T2DM and AC from the inception of these search services up to pre-sent. Results: We detected a total of 435 eligible records. After we applied the inclusion and ex-clusion criteria, a total of 14 articles were included in the present systematic review. Included manuscripts focused on the potential role of T2DM as a risk factor for the development of AC and on its contribution to a worse prognosis in AC, e.g., development of sepsis or other complications, the risk of AC recurrence and the impact on mortality. Conclusions: As compared to non-diabetic individuals, patients with T2DM have a higher risk of AC as complication of choledocholithiasis or gallstone pancreatitis. Several oral hypoglycemic drugs used in the management of T2DM may also be involved in the onset of AC. Diabetic patients who suffer from AC have a higher likelihood of longer hospital stays and sepsis, as well as a higher risk of mortality and more severe forms of AC as compared to non-diabetic individuals.

 Methods:

-          “Full-text access was possible for the included studies” should not be an inclusion/exclusion criterion. Please specify how many full-text you excluded as these cannot be retrieved and the specific reasons why those full-text could not be retrieved.

Response: Thank you for your valuable comment. We have corrected this section of the manuscript. In fact, we did not exclude any full-texts because all were available.  

Results:

-          The results section results are poor. No analysis has been carried out. Data has not been summarised in the results sections. It should be definitively edited. Some findings are described in the discussion sections “T2DM was associated with a longer hospital stay, sepsis, higher risk of mortality, and more severe AC as compared to non-diabetic patients”. However, these findings are not supported by any analytical effort provided by the authors.

Response: Thank you for your valuable suggestion. We have revised the Results section. However, we must point out that a quantitative analysis was not possible because most of the results were derived from single studies. Thus, only a qualitative synthesis of the results was possible.       

The 14 selected manuscripts were published between 2011 and 2022 [27-40]. Most of the analyzed data was derived from retrospective cohort studies [27-35],[37-40] with the exception of two investigations, namely one prospective study [36] and one case se-ries [34]. The number of subjects who partook in these assessments ranged from 12 to 123 990 [27-40]. Most of the research was carried out in Asia: two studies were conduct-ed in Singapore [28,36], two in China [31-32], one in India [37], one in South Korea [35] and one in Israel [33]. The remaining studies were conducted in the United States of America, i.e., two studies [27,38], in Europe, i.e., Denmark [40], France [34] or Spain [30], or were the fruit of international collaboration [39]. Most manuscripts focused on the potential role of DM as a risk factor for the development of AC or on its contribution to a worse prognosis (e.g., sepsis, recurrence) or to an elevated mortality in AC [27-33], [35-36], [38,40]. Two studies focused on the etiology of AC and sought to depict an asso-ciation between the microorganisms involved and the presence of DM [34,37]. One as-sessment pointed out an association between antidiabetic drugs and the risk of AC [39]. The main particularities identified in this systematic review are summarized in Table 1.

Overall, T2DM increased the risk of AC onset in patients with choledocholithiasis, gallstone pancreatitis, patients who underwent ERCP, and subjects with T2DM who were treated with oral hypoglycemic agents. T2DM was associated with longer hospital stay, sepsis, higher risk of mortality, and more severe forms of AC as compared to non-diabetic individuals [27-40].

 Conclusions:

-          Conclusions are not supported by data provided by the present paper.

Response: Thank you for your valuable suggestion. We have revised the conclusions accordingly.

As compared to non-diabetic individuals, patients with T2DM have a higher risk of AC as a complication of choledocholithiasis, gallstone pancreatitis, or ERCP. Some oral hypoglycemic drugs used in the management of T2DM may also cause AC. Dia-betic patients suffering from AC have a higher likelihood of a longer hospital stay, sep-sis, elevated risk of mortality, and more severe forms of AC as compared to non-diabetic individuals. Hence, this systematic review found an association between T2DM and worse outcomes in patients suffering from AC. Statistical significance of these find-ings may be found by performing quantitative analysis in future studies. Given the small number and the limitations of the studies included in this systematic review, the authors recommend further large prospective studies are needed to investigate how, not only T2DM as a whole but, more importantly, its duration, glycemic control and other components of the metabolic syndrome influence the prognosis of patients with AC. A comparison or control arm and the use of the 2018 TG for risk and severity strat-ification criteria for AC should also be used to further increase the quality of the results. Last but not least, it is equally important to identify which are the best management options for these particular patients in order to achieve lower mortality and incidence of complications. Quantitative analysis may provide a stronger scientific value to papers as compared to qualitative analysis conducted in systematic reviews. 4. A signalling figure as per the title and conclusion of your manuscript is needed to improve the quality of the manuscript.

Minor: There are some typos throughout the text (Web of Science in fig. 1)

Response: Thank you for your valuable suggestion.

Thank you for your valuable suggestions. We do hope you will find the revised version of the paper significantly improved and worthy of publication in Healthcare.

Yours sincerely,

The authors.

Round 2

Reviewer 2 Report

Dear Authors,

I have read again your manuscript with great interest. I believe its quality has been improved following reviewers' suggestions.